# EXPLORE AND CONTROL WITH ADVERSARIAL SURPRISE

## ABSTRACT

Unsupervised reinforcement learning (RL) studies how to leverage environment statistics to learn useful behaviors without the cost of reward engineering. However, a central challenge in unsupervised RL is to extract behaviors that meaningfully affect the world and cover the range of possible outcomes, without getting distracted by inherently unpredictable, uncontrollable, and stochastic elements in the environment. To this end, we propose an unsupervised RL method designed for high-dimensional, stochastic environments based on an adversarial game between two policies (which we call Explore and Control) controlling a single body and competing over the amount of observation entropy the agent experiences. The Explore agent seeks out states that maximally surprise the Control agent, which in turn aims to minimize surprise, and thereby manipulate the environment to return to familiar and predictable states. The competition between these two policies drives them to seek out increasingly surprising parts of the environment while learning to gain mastery over them. We show formally that the resulting algorithm maximizes coverage of the underlying state in block MDPs with stochastic observations, providing theoretical backing to our hypothesis that this procedure avoids uncontrollable and stochastic distractions. Our experiments further demonstrate that Adversarial Surprise leads to the emergence of complex and meaningful skills, and outperforms state-of-the-art unsupervised reinforcement learning methods in terms of both exploration and zero-shot transfer to downstream tasks.

## 1 INTRODUCTION

Reinforcement learning methods have attained impressive results across a number of domains (e.g., Berner et al. (2019); Kober et al. (2013); Levine et al. (2016); Vinyals et al. (2019)). However, current RL methods typically require a large number of samples for each new task (Dann et al., 2018). In other areas of machine learning, an effective way to mitigate high data requirements has been the use of unsupervised or self-supervised learning (Sutskever et al., 2014; Radford et al., 2019). Similarly, humans and animals seem to be able to learn rich priors from their own experience without being told what to do, and children engage in structured but unsupervised play in part as a way to acquire a functional understanding of the world (Smith & Gasser, 2005). Based on this intuition, unsupervised RL methods rely on intrinsic motivation (IM): task-agnostic objectives that incentivize the agent to autonomously explore the world and learn behaviors that can be used to solve a range of downstream tasks with little supervision. A general strategy is to exactly or approximately express this task-agnostic objective in the form of a reward function that uses only environment statistics, and then optimize it using standard RL algorithms.

A reasonable goal for a good unsupervised learning algorithm is to fully explore the state space of the environment, since this ensures the agent will have the experience which with to learn the optimal policy for a downstream task. Therefore, past work on IM has frequently focused on novelty-seeking agents that maximize surprise or prediction error (Achiam & Sastry, 2017; Schmidhuber, 1991; Yamamoto & Ishikawa, 2010; Pathak et al., 2017; Burda et al., 2018). However, these methods are vulnerable to becoming distracted by inherently stochastic elements of the environment, such as a *"noisy TV"* (Schmidhuber, 2010). In contrast, active inference researchers inspired by biological agents have focused on developing agents that seek to control their environment and *minimize* surprise (Friston, 2009; Friston et al., 2009; 2016; Berseth et al., 2021). These methods suffer from the opposite issue, the *"dark room problem"*, in which a surprise-minimizing agent in a low-entropy

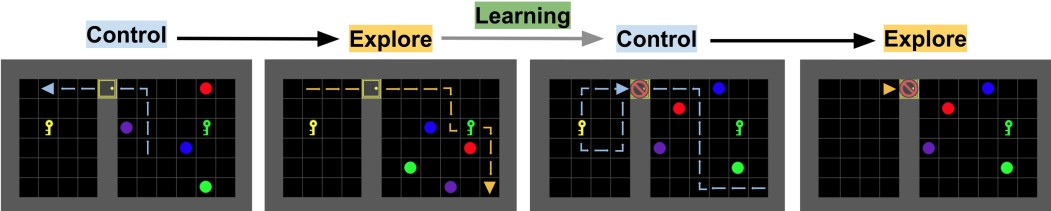

Figure 1: Adversarial Surprise is a multi-agent competition in which two policies take turns controlling a single agent. The Explore policy acts first, and tries to put the agent into surprising, high entropy states. On its turn, the Control policy tries to minimize surprise by finding familiar, low-entropy, predictable states. As training continues, the competition drives the agents to learn increasingly complex behaviors. In the above example, the Control policy eventually learns to pick up a key and lock the door to prevent the Explore agent from taking it into the room with randomly moving objects (a *noisy TV* state).

environment does not need to learn any behaviors at all in order to satisfy its objective (Friston et al., 2012). Yet humans seem to maintain a balance between optimizing for both novelty and familiarity. For example, a child in a play room does not just try to toss their toys on the floor in every possible pattern, or immediately put them away in the toy box, but instead tries to stack them together, find new uses for parts, or combine them in various structured ways.

We argue that an effective unsupervised RL method should find the right balance between exploration and control. With this goal in mind, we introduce a new algorithm based on an adversarial game between two policies, which take turns sequentially acting for the same RL agent. The goal of the *Control* policy is to minimize surprise, by learning to manipulate its environment in order to return to safe and predictable states. In turn, the *Explore* policy is novelty-seeking, and attempts to maximize surprise for the Control policy, putting it into a diverse range of novel states. When combined, the two adversaries engage in an arms race, repeatedly putting the agent into challenging new situations, then attempting to gain control of those situations. Figure 8 shows an illustration of the method, including a sample interaction. Rather than simply adding noise to the environment, the Explore policy learns to adapt to the Control policy, and to search for increasingly challenging situations from which the Control policy must recover. Thus our method, *Adversarial Surprise* (AS), leverages the power of multi-agent training to generate a curriculum of increasingly challenging exploration and control problems, leading to the emergence of complex, meaningful behaviours.

The contributions of this paper are: i) The Adversarial Surprise algorithm; ii) Theoretical results which prove that when the environment is formulated as a stochastic Block MDP (Du et al., 2019), traditional surprise-maximizing methods will fail to fully explore the underlying state space, but Adversarial Surprise will succeed; iii) Empirical evidence which supports the theoretical results, and shows that AS fully explores the state space of both Block MDPs and traditional benchmarking environments like VizDoom; iv) Experiments that compare AS to state-of-the-art unsupervised RL baselines Random Network Distillation (RND) (Burda et al., 2018), Asymmetric Self-Play (ASP) (Sukhbaatar et al., 2017), Adversarially Guided Actor-Critic (AGAC) (Flet-Berliac et al., 2021), and Surprise Minimizing RL (SMiRL) (Berseth et al., 2021), and show that AS is able to explore more effectively, learn more meaningful behaviors, and achieve higher task reward when transferred zero-shot to common benchmarking environments Atari and VizDoom. Videos of our agents are available on the project website: `https://sites.google.com/corp/view/adversarialsurprise/home`, and show that AS is able to learn interesting, emergent behaviors in Atari, VizDoom, and MiniGrid, even *without ever having trained with the game reward*.

## 2    RELATED WORK

**Novelty-seeking and exploration methods** lead the agent to increase coverage of the environment. A simple way to implement novelty-seeking is to maximize the prediction error of a world model (Achiam & Sastry, 2017; Schmidhuber, 1991; Yamamoto & Ishikawa, 2010; Pathak et al., 2017; Burda et al., 2018; Raileanu & Rocktäschel, 2020; Zhang et al., 2020b). Random Network Distillation (RND) (Burda et al., 2018) is a highly effective example of one such method. However, a major problem of prediction-error-based methods is that the intrinsic reward is not useful when the envi-

ronment contains aleatoric uncertainty, i.e. inherently stochastic elements. This problem is often referred to as the *noisy TV problem*, after Schmidhuber (2010) used the example of an agent becoming stuck staring at static on a TV screen. We show in this work that RND is indeed vulnerable to this problem, performing poorly in stochastic environments.

Several works have proposed solutions to deal with aleatoric uncertainty. For example, some approximate information gain on the agent's dynamics model of the environment (Houthooft et al., 2016; Still & Precup, 2012; Bellemare et al., 2016; Pathak et al., 2017; Schmidhuber, 1991), using variational Bayes or ensemble disagreement (Shyam et al., 2019; Pathak et al., 2019; Houthooft et al., 2016). However, implementing such Bayesian procedures is difficult, because it requires scalable and effective modeling of epistemic uncertainty, which itself is a major open problem with high-dimensional models such as neural networks (Bhattacharya & Maiti, 2020). Another method based on maximizing information gain between the agent's actions and future state is known as *empowerment* (Klyubin et al., 2005; Salge et al., 2014; Eysenbach et al., 2018; Sharma et al., 2019), but can also be difficult to approximate for high-dimensional states (Karl et al., 2015; Zhao et al., 2020; de Abril & Kanai, 2018; Zhang et al., 2020a; Mohamed & Rezende, 2015; Gregor et al., 2016; Hansen et al., 2019). Instead of attempting to directly approximate information gain, our approach maximizes state coverage via an adversarial competition over observation entropy. As we will show, this recovers an effective coverage strategy even in the presence of rich, stochastic observations, and performs well in practice.

The asymptotic policy learned by standard novelty-seeking methods is not exploratory. Recent work tries to learn a policy that approximately maximizes the state marginal entropy at convergence (Hazan et al., 2019; Lee et al., 2019). The state marginal entropy is hard to compute in general, and recent work has proposed various approximations (Seo et al., 2021; Liu & Abbeel, 2021; Mutti et al., 2021). We prove that even with only noisy observations of the underlying state, our method asymptotically maximizes the state marginal entropy at convergence under some assumptions.

**Surprise minimization and active inference:** The design of the Control agent in our method draws on ideas from surprise minimization and active inference (Friston, 2009; Friston et al., 2009; 2016). The free energy principle, originating in the neuroscience community, argues that complex niche-seeking behaviors of biological systems are the result of minimizing long-term average surprise on system sensors, leading agents to stay in safe and stable states (Friston, 2009). SMiRL is an unsupervised RL method that leverages surprise minimization as a means to discover skills that stabilize an otherwise chaotic environment (Berseth et al., 2021). However, such approaches require strong assumptions on the stochasticity of the environment. In low-entropy environments, surprise minimization will not lead to learning interesting behavior due to the *dark room problem*, in which the agent is not incentivized to explore the environment to find a better niche (Friston et al., 2012). Our method does not require that the environment is stochastic, since the Explore agent itself drives the Control agent into situations from which surprise minimization is challenging.

**Multi-Agent competition** has been shown to drive the emergence of complex behavior (Baker et al., 2019; Dennis et al., 2020; Xu et al., 2020; Leibo et al., 2019; Schmidhuber, 1997; Campero et al., 2020). Asymmetric Self-Play (ASP) aims to learn increasingly complex skills via a competition between two policies, where one policy (Bob) tries to imitate or reverse the trajectory of the other policy (Alice) (Sukhbaatar et al., 2017; OpenAI et al., 2021). Our empirical results compare to ASP, and demonstrate that it can fail in stochastic environments, because Alice can easily produce random trajectories which are very difficult for Bob to imitate. Similar to ASP, Adversarially-Guided Actor Critic (AGAC) (Flet-Berliac et al., 2021) introduces an adversary agent which attempts to mimic the action distribution of the actor, while the actor tries to differentiate itself from the adversary. Like these methods, Adversarial Surprise uses a two-player game to induce exploration and emergent complexity. However, our objective is general and information-theoretic, focusing on minimization or maximization of surprise rather than reaching specific states. Unlike these methods, we provide theoretical results showing AS provides a principled approach to maximizing state coverage in stochastic environments. Finally, our method is reminiscent of recent work that learns separate exploration and exploitation policies but without competition between them (Badia et al., 2020b;a; Campos et al., 2021)

## 3 BACKGROUND

**Markov Decision Process (MDP):** An MDP is a tuple $(\mathcal{S}, \mathcal{A}, \mathcal{T}, r, \gamma)$, where $s \in \mathcal{S}$ are states, $a \in \mathcal{A}$ are actions, $r(a, s)$ is the reward function, and $\gamma \in [0, 1)$ is a discount factor. At each timestep $t$, an RL agent selects an action $a_t$ according to its policy $\pi(a_t | s_t)$, receives reward $r(a_t, s_t)$, and the environment transitions to the next state according to $\mathcal{T}(s_{t+1} | s_t, a_t)$. The agent uses RL to maximize its cumulative discounted reward over the episode: $R = \sum_{t=0}^{T} \gamma^t r^i(a_t, s_t)$.

**Block MDP (BMDP):** A BMDP (Du et al., 2019) extends the MDP formalism to the case where the agent cannot observe the true state $s$, but rather observes rich observations $o \sim p(O|s)$. Unlike a traditional partially-observed MDP (POMDP), a BMDP has an additional disjointness assumption: for any $s, s' \in S$, $s \neq s' \Rightarrow \text{supp}(p(O|s)) \cap \text{supp}(p(O|s)) = \emptyset$. The same assumption has been used in several prior works (Azizzadenesheli et al., 2016; Dann et al., 2018; Krishnamurthy et al., 2016; Misra et al., 2020), and enables the BMDP to naturally capture a wide range of environments which have rich, high-dimensional observations. We are interested in stochastic BMDPs, in which the observation distribution is inherently entropic for some states, i.e. $\exists s : H(p(O|s)) > 0$, where $H$ is the entropy. This is a conceptual model of stochastic environments with uncontrollable factors.

**Maximizing state marginal entropy:** The state marginal distribution $d^\pi(s)$ of a policy $\pi$ is the probability that the policy visits state $s$: $d^\pi(s) = \mathbb{E}_{a \sim \pi, s \sim \mathcal{T}}[\frac{1}{T} \sum_{t=0}^{T} \gamma^t \mathbb{1}(s_t = s)]$ (Lee et al., 2019). The *observation* marginal density is defined analogously:

$$d^\pi(o) = \mathbb{E}_{a \sim \pi, s \sim \mathcal{T}, o \sim p(\mathcal{O}|s)} \left[ \frac{1}{T} \sum_{t=0}^{T} \gamma^t \mathbb{1}(o_t = o) \right] \tag{1}$$

We are interested in agents that are able to fully explore the underlying state space of a stochastic BMDP; that is, agents that can maximize the entropy of the state marginal distribution: $H(d^\pi(s))$. However, in the context of a BMDP, agents cannot directly observe the underlying state $s$ and must attempt to maximize $H(d^\pi(s))$ purely through their ability to alter $d^\pi(o)$.

## 4 ADVERSARIAL SURPRISE

The design of our method is guided by the following desiderata: i) It should be able to learn meaningful behaviors even if the environment has high-dimensional, stochastic observations, and avoid the noisy TV problem. Thus, our method should favour low-entropy states when possible, while still seeking novelty. ii) It should enable the emergence of skills. Since multi-agent training can lead to the emergence of complex skills, we create a multi-agent adversarial game between two policies. iii) The optimized objective should be theoretically sound, in the sense that we can provably show that it optimizes a well-defined exploration metric under some assumptions. In our case, we will show that our two-player game maximizes coverage of the underlying state space of a stochastic BMDP.

To achieve these goals, we design an adversarial game that pits two policies against each other in a competition over the amount of surprise encountered through exploration. Specifically, we learn an Explore policy, $\pi^E$, and a Control policy, $\pi^C$. The goal of the Control policy is to minimize its own surprise, or observation entropy, using a learned density model $p_\theta(o)$. The Explore policy's goal is to maximize the surprise that the Control policy experiences. The policies take turns taking actions for the agent, switching back and forth throughout the episode. To simplify notation, assume the policy taking actions switches every $k$ steps, such that for the first $k$ steps, $a_t \sim \pi^E(a_t|o_t)$ (the Explore policy acts), then for timesteps $k - 2k$, $a_t \sim \pi^C(a_t|o_t)$ (the Control policy acts). The policies continue switching until the end of the episode. Each policy is given $k$ steps to act to enable it to reach states that will be challenging for the other policy to recover from, thus facilitating learning more complex and long-term exploration and control behaviors (see Figure 8).

Surprise is computed using a learned density model that estimates the agent's likelihood of experiencing observation $o$, $p_\theta(o)$. Following Berseth et al. (2021), $p_\theta(o)$ is re-initialized each episode and trained using maximum likelihood estimation (MLE) to fit the observations of the agent within a single episode, which are stored in a buffer $\beta$. The density model is either represented using a multivariate Gaussian distribution fit to the pixels within $o$ (as in Berseth et al. (2021)), or using independent categorical distributions for each pixel. For more details see Appendix C.

Because the Control policy is surprise-minimizing, its reward is $r_t^C = \log p_\theta(o_t)$, which resembles SMiRL (Berseth et al., 2021), except using the observation in place of the state. The goal of the Explore policy is to *maximize* the observation surprise *during the Control agent's turn*. This creates an adversarial game, in which the Explore policy attempts to find surprising situations with which to expose the Control policy, and the Control policy's objective is to recover from them. Therefore, the Explore policy's reward is based on the surprise for the observations of the Control policy. Assume that the Control policy's turn begins at timestep $t^C$, and it receives a total reward of $R^i = \sum_{t=t^C}^{t^C+k} \gamma^k \log p_\theta(o_t)$ for that turn. Then, the Explore policy's reward is $-R^i$, and is applied to the last timestep of the Explore policy's turn (i.e. timestep $t^C - 1$). Thus, Adversarial Surprise defines the following adversarial game between the two policies:

$$\max_{\pi^E} \min_{\pi^C} -\mathbb{E}\left[\sum_{t=t^C}^{t^C+k} \log p_\theta(o_t)\right], \tag{2}$$

where the Explore policy can only effect $p_\theta(o_t)$ through the final state that it produces at the end of its turn, which is the initial state for the Control policy. Algorithm 1 (Appendix D) gives the full training procedure.

We can show that optimizing these rewards leads the two players to compete over the amount of observation entropy, where the Explore policy's objective is to maximize observation entropy:

$$\mathcal{J}^E = -\mathbb{E}\left[\sum_{t=t^C}^{t^C+k} \log p_\theta(o_t)\right] \approx H(d^{\pi^C}(o)), \tag{3}$$

And the Control policy's goal is to minimize its observation entropy: $-H(d^{\pi^C}(o))$. We prove the relationship between $\log p_\theta(o)$ and the observation entropy in Appendix A.

**Implementation details:** We parameterize the policies for the Explore and Control policy using deep neural networks (NN) with parameters $\phi^E$ and $\phi^C$, respectively. The policy is based on a convolutional NN, which takes in the last 4 observation frames. The networks are trained using Proximal Policy Optimization (PPO) (Schulman et al., 2017); further details are given in Appendix C. We have found it helpful to only compute the surprise reward using observations from the second half of the Control policy's turn. This gives the agent greater ability to take actions that may lead to initial surprise, but reduce entropy over the long term. Finally, we allow the Explore and Control policys to act for a different number of timesteps (that is, we have a separate $k^C$ and $k^E$), enabling us to tune the emphasis on exploration or control depending on the environment.

### 4.1 Adversarial Surprise maximizes state coverage in Block MDPs

We will show that AS covers the underlying state space of a Block MDP, under some assumptions about the density of states with low observation entropy. Full proofs are in Appendix A.

We are interested in maximizing the entropy of the marginal *state* distribution $H(d^\pi(s))$, using a density model of the observation, $p_\theta(o)$. To that end, we prove the following lemma in Appendix A:

**Lemma 1.** *The cumulative surprise measured by the observation density model $p_\theta(o)$ forms an upper bound of the observation marginal entropy $H(d^\pi(o))$, which becomes tight when the observation density model fits the observation marginal $d^\pi(o)$: $-\mathbb{E}_\pi \sum_{t=0}^\infty \log p_\theta(o_t) \geq H(d^\pi(o))$.*

**Lemma 2.** *Given the disjointness assumption of the BMDP stated in Section 3, we show a useful relation between observation marginal entropy and state marginal entropy:*

$$H(d^\pi(o)) = \mathbb{E}_{d^\pi(s)} H(p(O|S = s)) + H(d^\pi(s)). \tag{4}$$

See Appendix A for the proof. Equation 4 shows that maximizing entropy in the marginal observation distribution $d^\pi(o)$ amounts to maximizing two terms: the emission entropy $H(p(O|S))$, and the state marginal entropy $H(d^\pi(s))$.

Denote by $\mu_{RND}^*$ the state stationary distribution of an RND agent trying to maximize the observation stationary distribution. It follows by lemma 2 that $\mu_{RND}^*$ is the solution to the entropy-regularized LP

$$\mu_{RND}^* = \arg\max_\mu \langle \mu, h \rangle + H(\mu)$$

where $h$ is the vector indexed by $S$ giving the emission entropy of all states. This program has a closed form solution which is given by $\mu^*_{RND\,i} = \frac{e^{h_i}}{\sum_j e^{h_j}}$. Therefore, the distribution exponentially favors states with high emission entropy. This explains why algorithms which maximize observation entropy fail to explore environments with noisy TV states.

In Adversarial Surprise, the Controller policy is actively trying to transform the observation distribution given by the Explorer policy into a low entropic distribution after $T$ steps. Therefore the objective can be written as

$$\min_M \max_\mu \langle M\mu, h \rangle + H(M\mu)$$

where $M$ must lie in $\{P^T_\pi : \pi \text{ is a stationary policy}\}$. Therefore the Explorer policy is now constrained to find a maximum in the image of $M$. We provide now a sufficient condition on the structure of the BMDP such that the Control policy can choose a matrix $M$ that induces the Explore policy to maximize the state entropy by maximizing the observation entropy after $T$ steps.

We first define a (semiquasi)metric on the latent state space: $\tilde{d}(s, s') = \min\{k : \exists \pi, P^\pi_k(s'|s) = 1\}$, where by convention $P^\pi_0(s|s) = 1$ for all $s$. In other words, $\tilde{d}(s, s') = k$ if there is a policy that reaches $s'$ from $s$ in $k$ steps with probability 1. We symmetrize this metric by defining the following semimetric: $d(s, s') = \max\{\tilde{d}(s, s'), \tilde{d}(s', s)\}$

**Definition 1.** *We now give a formal definition of "dark rooms". We say that a state $s$ is a **dark room** if it has minimal emission entropy: $H(p(O|S = s)) = \min_{s \in S} H(p(O|S = s))$.*

**Assumption 1.** *We assume that for every state $s$, there is a dark room $s'$ such that $d(s, s') = T$. That is, the set of dark rooms is a $T$-cover of the state space with respect to $d$, and from every state there is a dark room can be reached in exactly $T$ steps.*

We can now state our main result:

**Theorem 1.** *Under the BMDP assumption and Assumption 1, the Markov chain induced by the following AS game:*

$$\max_{d^{\pi_E}(s_0)} \min_{d^{\pi_C}_{1:T}(s|s_0)} H\left(d^{\pi_C}_T(o)\right)$$

*$T$-covers the latent state space, i.e., for all states $s$, there is a state $s'$ such that $d^\pi(s') > 0$ and $d(s, s') \leq T$, where $d^\pi$ is the state marginal distribution induced by the game between the Explore ($\pi_E$) and Control ($\pi_C$) policies.*

*Proof.* (sketch.) Assumption 1 guarantees that for any states that the Explore policy reaches, the Control policy can find a low-emission-entropy state within its turn ($T$ timesteps), so that $H(p(O|s))$ is minimized. Thus, by Lemma 2, when the Control policy is factored out, the objective of the Explore policy becomes:

$$\mathcal{J}^E = H(d^\pi(o)) \approx H(d^\pi(s)). \tag{5}$$

Therefore the Explore policy's objective is to maximize coverage of the BMDP state space.  □

## 5 EXPERIMENTAL RESULTS

We now present experimental results[1] designed to assess the following questions:

**Q1. State coverage:** How well does AS explore the underlying state space in a stochastic BMDP, as compared to baselines? Given the theoretical results in Section 4.1, we hypothesize that AS will fully explore the environment, but other methods will become distracted by stochastic elements.

**Q2. Zero-shot transfer:** Does AS learn behaviors that are useful for downstream tasks? To assess this, we train AS and baseline methods using only intrinsic reward, then transfer the agents zero-shot to standard benchmark environments in Atari (Bellemare et al., 2013) and VizDoom (Kempka et al., 2016b), and measure the amount of game reward obtained. While there is no reason to expect AS to always correlate with the objectives in arbitrary MDPs, we expect that the twin goals of maximizing

---

[1]AS performance is measured across the full episode, including both the Explore and Control turns.

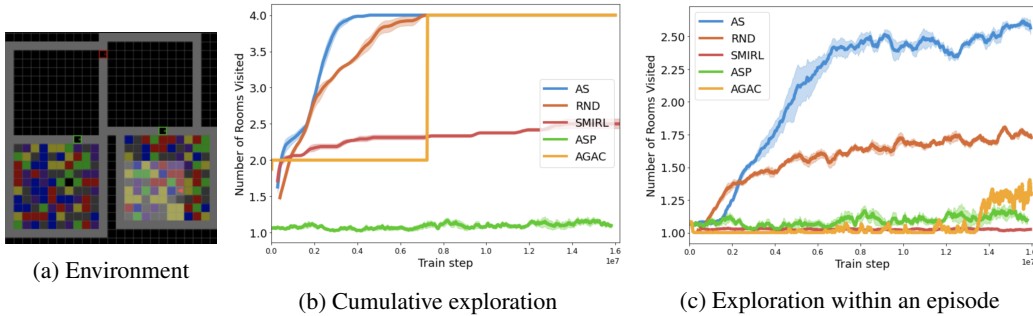

(a) Environment

(b) Cumulative exploration

(c) Exploration within an episode

Figure 2: **State coverage in stochastic BMDPs:** the number of rooms (out of 4) explored both (b) cumulatively over training, and (c) within an episode. AS explores better than the state-of-the-art exploration methods RND and ASP, which become distracted by noisy elements. AS outperforms SMiRL, which minimizes observation surprise by turning to face the wall, or remains in dark rooms.

coverage while achieving high control should correlate well with objectives in many reasonable MDPs. This is particularly true of games, which have a notion of progress that roughly corresponds to coverage, but at the same time have many dangerous states that could result in 'death', which leads to an unexpected jump back to the starting state. We hypothesize that AS should, without even being aware of the task reward, perform well in these environments. Comparing to prior methods in these domains is interesting, because prior work has variously argued that both novelty-seeking exploration methods (Burda et al., 2018) and surprise-minimization methods (Berseth et al., 2021) should be expected to achieve high scores in these games.

**Q3. Emergence of complexity:** If AS is able to achieve high zero-shot transfer performance, we hypothesize that it will be because the adversarial game drives the acquisition of increasingly complex observable behaviors. Therefore, we track the emergence of specific skills throughout training, and qualitatively examine final performance in benchmark environments to identify meaningful skills.

**Baselines:** We compare AS to four competitive unsupervised RL baselines: i) Asymmetric Self-Play (ASP) (Sukhbaatar et al., 2017), a state-of-the-art multi-agent curriculum method; ii) Random Network Distillation (RND) (Burda et al., 2018), a state-of-the-art exploration method; iii) Adversarially Guided Actor-Critic (AGAC) (Flet-Berliac et al., 2021), a recently proposed adversarial exploration method; and iv) Surprise Minimizing RL (SMiRL) (Berseth et al., 2021), a recently proposed intrinsic motivation based on active inference. All methods use the same PPO implementation, with hyperparameters given in Appendix C.

**Environments:** To evaluate the above hypotheses, we use three types of environments. To obtain environments that match the precise specifications of the stochastic **BMDP** formalism, and present an exploration challenge, we constructed a custom family of procedurally generated navigation tasks based on MiniGrid (Chevalier-Boisvert et al., 2018). These environments contain rooms that are either empty (dark), or contain stochastic elements such as flashing lights that randomly change color. They also contain elements such as doors that can be opened with keys, and switches that, when flipped, stop or start the stochastic elements moving. As in MiniGrid, the agent only sees a 5x5 window of the true state. Each room of size 12 represents a hard exploration task for the agent (see Figure 2a).While gridworlds allow us to easily interpret the results and behaviors of the agent, and were used by ASP, SMiRL, and AGAC, we also compare to prior work on two standardized benchmarks with high-dimensional observations. Specifically, we use **Atari** ALE (Bellemare et al., 2013), which was used by both SMiRL and RND to establish their effectiveness, and the **ViZDoom** environment (Kempka et al., 2016b), which was used by AGAC and SMiRL. Due to limited computational resources, we do not conduct experiments in all possible Atari games (which is consistent with prior work (Berseth et al., 2021; Burda et al., 2018)), but we do not cherry-pick results; we show results for each of the games that we test.

**Q1: State coverage:** We measure state coverage in the procedurally-generated BMDPs using the number of rooms the agent visits (since many different observations can be generated from a single room). The results for all algorithms are shown in Figure 2. We measure coverage cumulatively, over the course of training (Figure 2b), to assess whether each method will lead the agent to collect experience from all possible states. This measure is relevant to whether the technique can be used

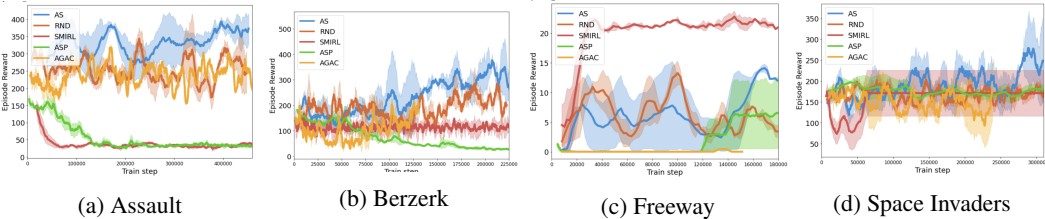

| (a) Assault | (b) Berzerk | (c) Freeway | (d) Space Invaders |

Figure 4: **Zero-shot transfer in Atari:** Each method is trained in Atari using only intrinsic reward, then transferred zero-shot to optimizing game reward. Plots show game reward, where error bars are the 95% Confidence Interval (CI) of 5 seeds. The games reward behavior such as staying alive and shooting enemies, so obtaining higher reward indicates the agent has learned meaningful behaviors. Across 3/4 environments, AS outperforms RND, ASP, AGAC, and SMiRL, showing AS provides a general way to learn useful behaviors in the absence of external reward.

as an effective exploration bonus to aid learning a downstream task. We also measure the number of rooms explored within each episode (Figure 2c). This allows us to assess whether the asymptotic policy learned by the algorithms continues to explore once it has converged.

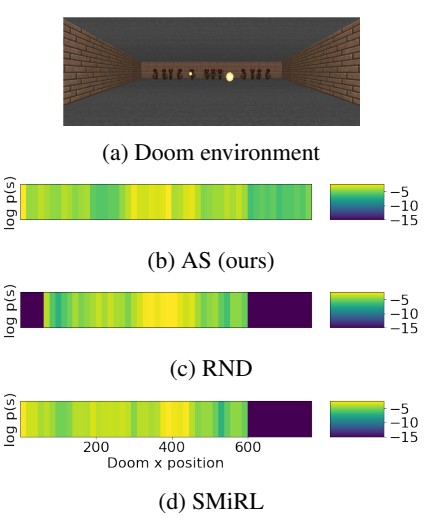

(a) Doom environment

(b) AS (ours)

(c) RND

(d) SMiRL

Figure 3: **State coverage in Doom:** the heatmaps show $\log p(s)$ over the first 1000 training steps, where $s$ is the agent's $x$-axis position in Doom. Black areas indicate $p(s) = 0$, meaning the agent has never visited these coordinates. These results show that AS fully explores the state space, while RND fails to explore both edges and SMiRL fails to explore the right edge.

As predicted by our theoretical analysis, we see that AS learns to more fully explore the environments than prior work, visiting all possible rooms over a lifetime (Figure 2b) and significantly more rooms per episode (Figure 2c). It learns more quickly and explores more thoroughly than RND, which becomes distracted by stochastic elements that lead to high prediction error. Stochasticity also hinders learning for ASP, since Alice can easily produce random goals that are difficult for Bob to replicate. Finally, we see that SMiRL, which is designed for fully observed environments, does not explore effectively because it suffers from the dark room problem – it prefers to stay within the empty rooms, and not venture into rooms with high-entropy, stochastic elements.

Figure 3 shows coverage of the latent state space in the VizDoom *Take Cover* environment as positions along the x-axis. In this environment, agents navigate horizontally while seeing a partial view of the underlying state. Figure 3 reveals that only AS covers all the positions in the map in the first 1000 train steps, whereas RND fails to explore the edges, and SMiRL fails to explore the right edge.

**Q2. Zero-shot transfer:** Figures 4 and 5 show how each method performs when transferred zero-shot to the task of optimizing game reward in several Atari environments and VizDoom. Because the game rewards complex behaviors like shooting or avoiding enemies, a high game reward indicates the agent has learned interesting skills, purely from optimizing the intrinsic objective. Across the environments, AS performs better than RND, SMiRL, AGAC, and ASP. While RND is sometimes effective, its performance often decreases over time due the bonus from the prediction error shrinking as more states become familiar. Further, maximizing novelty in environments like Freeway, Space Invaders, and Doom can lead to the agent dying, corresponding to low reward. SMiRL performs best in Freeway, where minimizing entropy corresponds closely to staying alive and not being hit by cars. However, SMiRL performs poorly in the other environments because it avoids entropy by hiding from enemies (staying in dark rooms when they are available). ASP also performs poorly because it is possible for Alice to quickly reach states which Bob cannot easily replicate, preventing the algorithm from learning meaningful behaviors. For AGAC, the adversary cannot accurately predict the actions of the agent given an observation when the observation has a high entropy emission probability. In contrast, AS consistently obtains high returns across all environments, indicating that optimizing for both explo-

ration and control provides a broadly useful inductive bias for learning interesting behaviors in the absence of external reward.

**Q3.** **Emergence of complexity:** Due to space constraints, we show results for the emergence of a series of interesting behaviors in the BMDP environments in Appendix B. Figure 7 reveals that the multi-agent game induced by AS leads to a curriculum with multiple training phases: the Control agent first learns to go to a dark room, the Explore agent learns to go back to a noisy TV, and the Control agent responds by learning to *lock a door* to make it more difficult for the Explore agent to expose it to surprising states. This highlights the potential of Adversarial Surprise to learn *long-term* surprise-minimizing behaviors. Figure 6 demonstrates that RND and ASP do not learn to try taking even simple actions to control the environment, such as flipping a switch to stop stochastic elements from moving. In AGAC, the actor chooses actions that maximize the discrepancy between its policy and the Adversary's. Although this proves to be effective for exploration between subsequent trajectories, as shown in Figure 5, the interaction between the Control and Ex-

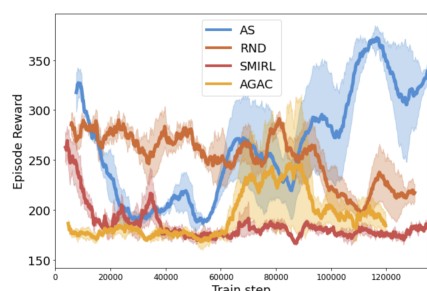

Figure 5: **Zero-shot transfer in Doom.** Consistent with the Atari results, AS learns more meaningful behaviors (i.e. moving while avoiding enemy bullets) than other techniques. This leads to higher environment reward.

plore policies in Adversarial Surprise is able to facilitate more complex behaviors by not only finding novel states to explore but determining when to exhibit control over safe states. The emergence of interesting behaviour in both Atari and Vizdoom is evidenced in the videos available at: `https://sites.google.com/corp/view/adversarialsurprise/home`. Purely through optimizing the Adversarial Surprise objective, AS agents learn complex behaviors such as hiding from enemies behind walls or barriers, shooting enemies in Space Invaders, Berzerk, Assault, and Doom, and hopping across the road in Freeway.

## 6 DISCUSSION

We proposed Adversarial Surprise as a general approach for unsupervised RL. Adversarial Surprise corresponds to a two-player adversarial game, in which two policies compete over the amount of surprise, or observation entropy, that an agent experiences. Reminiscent of Dr. Jekyll and Mr. Hyde, the Explore policy acts to expose the Control policy to highly entropic states from which it must recover by learning to manipulate the environment. We show that AS produces increasingly complex control and exploration strategies, and is able to address exploration in stochastic Block MDPs. In such environments, prior methods can become distracted by noisy elements, or suffer from the dark room problem.We show both theoretically and empirically that AS is robust against these issues, and learns to explore the environment more thoroughly, and control it more effectively, than state-of-the-art prior works like RND, ASP, AGAC, and SMiRL.

**Future work:** Our evaluation of AS focuses on coverage and unsupervised exploration, where we demonstrate that AS improves over novelty-seeking, surprise minimization, adversarial, and goal-setting methods in stochastic BMDPs and standard benchmarks like Atari and Doom. However, the potential value of unsupervised RL methods extends more broadly: such methods could be used to acquire skills for downstream task learning, controlling an environment to reach states from which more behaviors could be performed successfully. Future work could study how AS and its extensions could enhance applications like robotics, for example by collecting data for downstream reward-guided learning. Further, we see a potentially exciting method which combines AS with hierarchical RL, by training a meta-policy to select when to invoke the Explore and Control sub-policies. In this way, the meta-policy could explicitly decide when to explore and when to exploit.

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

## A  PROOF DETAILS

Firstly, we notice that we have a simple relation between marginal observation entropy and marginal state entropy by the structure of the POMDP:

$$d^\pi(o) = (1 - \gamma) \sum_t \gamma^t p(o_t = o) \tag{6}$$

$$= (1 - \gamma) \sum_t \gamma^t \sum_s p(s_t = s) p(o|s) \tag{7}$$

$$= (1 - \gamma) \sum_s p(o|s) \sum_t \gamma^t p(s_t = s) \tag{8}$$

$$= \sum_s p(o|s) d^\pi(s) \tag{9}$$

We can use this relation to prove the following lemma:

**Lemma 3.** *The cumulative surprise measured by the observation density model $p_\theta(o)$ forms an upper bound of the observation marginal entropy $H(d^\pi(o))$, which becomes tight when the observation density model fits the observation marginal $d^\pi(o)$.*

*Proof.*

$$-\mathbb{E}_\pi \log p_\theta(o) = -\sum_s d^\pi(s) \sum_o p(o|s) \log p_\theta(o) \tag{10}$$

$$= -\sum_o d^\pi(o) \log p_\theta(o) \tag{11}$$

$$\geq H(d^\pi(o)) \tag{12}$$

where the last inequality is because $-\sum_o d^\pi(o) \log p_\theta(o) - H(d^\pi(o)) = KL(d^\pi(o)||p_\theta(o)) \geq 0$  □

We suppose that we are in the Block MDP setting:

**Assumption 2** (Block MDP). *We suppose that for any $s, s' \in S$, $s \neq s' \Rightarrow supp(p(O|s)) \cap supp(p(O|s)) = \emptyset$*

In this case the marginal observation entropy can also be simply related to the marginal state entropy:

**Lemma 4.** *In a block MDP (BMDP) Misra et al. (2020), by noticing that $H(S|O) = 0$, we can decompose the observation marginal entropy as follows:*

$$H(d^\pi(o)) = \mathbb{E}_{d^\pi(s)} H(O|S = s) + H(d^\pi(s)) \tag{13}$$

*Proof.*

$$H(d^\pi(o)) = \sum_{o,s} p(o|s) d^\pi(s) \log(\sum_s p(o|s) d^\pi(s)) \tag{14}$$

$$= \sum_s d^\pi(s) \sum_{o \in \mathcal{O}_s} p(o|s) \log(p(o|s) d^\pi(s)) \tag{15}$$

$$= \sum_s d^\pi(s) [\sum_{o \in \mathcal{O}_s} p(o|s) \log p(o|s) + \log d^\pi(s)] \tag{16}$$

$$= \mathbb{E}_{d^\pi(s)} H(O|S = s) + H(d^\pi(s)) \tag{17}$$

□

that we can also obtain by simply noticing that $H(S|O) = 0$ in the Block MDP setting and writing the mutual information between $S$ and $O$.

Suppose that we have a small number of latent states with rich observations, that is, there is $s$ such that $H(O|S = s) \gg \log|\mathcal{S}|$. In this case, if we are trying to maximize the marginal observation entropy, we have:

$$\max_{d^\pi(s)} H(d^\pi(o)) \approx \max_{d^\pi(s)} \mathbb{E}_{d^\pi(s)} H(O|S = s) \tag{18}$$

The RHS is maximized by taking:

$$d^\pi(s) = I(s = \arg\max H(O|S = s)) \tag{19}$$

That is, we are stuck in a noisy TV.

On the contrary, if we are trying to minimize the marginal observation entropy, equation Eq.17 gives us the exact minimizer:

$$d^\pi(s) = I(s = \arg\min H(O|S = s)) \tag{20}$$

That is, we are stuck in a dark room.

Now suppose that we are optimizing the following objective:

$$\max_{d^{\pi_A}(s_0)} \min_{d^{\pi_B}_{1:T}(s|s_0)} H(d^{\pi_B}_{1:T}(o)) \tag{21}$$

**Definition 2.** *We define the following semiquasimetric in the state space:*

$$\tilde{d}(s, s') = \min\{k : \exists\pi, P^\pi_k(s'|s) = 1\} \tag{22}$$

where by convention $P^\pi_0(s|s) = 1$ for all $s$. In other words, $d(s, s') = k$ if state there is a policy that reaches $s'$ from $s$ in $k$ steps with probability 1.

**Definition 3.** *We define the following semimetric:*

$$d(s, s') = \max\{\tilde{d}(s, s'), \tilde{d}(s', s)\} \tag{23}$$

**Definition 4.** *We say that a state $s$ is a dark room if it has minimal emission entropy:*

$$H(O|S = s) = \min_{s \in S} H(O|S = s) \tag{24}$$

**Assumption 3.** *We make three assumptions concerning the density of dark rooms:*

(a) *We suppose that for every state $s$, there is a dark room such that $d(s, s') \leq T$. That is, the set of dark rooms is a $T$-cover of the state space with respect to $d$.*

(b) *We suppose that for every state $s$, there is a dark room such that $P^\pi_T(s'|s) = 1$, that is a dark room can be reached in exactly $T$ steps.*

(c) *We suppose that for any state $s$ and any dark room $s'$, if $\tilde{d}(s, s') \leq T$, then $d(s, s') \leq T$, that is if we can reach a dark room from a state $s$ in less than $T$ steps, then we can also reach $s$ from this dark room is less than $T$ steps.*

**Theorem 2.** *Under Assumptions 2 and 3, the Markov chain induced by the following AS game:*

$$\max_{d^{\pi_A}(s_0)} \min_{d^{\pi_B}_{1:T}(s|s_0)} H(d^{\pi_B}_T(o)) \tag{25}$$

*$T$-covers the state space, that is for every state $s$, there is a state $s'$ such that $d^\pi(s') > 0$ and $d(s, s') \leq T$, where $d^\pi$ is the marginal induced by the game between $A$ and $B$.*

*Proof.* Given an initial state $s_0$, Eq.17 shows that the controller will always reach the same state of lowest emission entropy at step $T$. By assumption 3(b) the Controller can always reach a dark room with probability 1 in exactly T steps. Therefore the game is equivalent to the following constrained objective:

$$\max_{d^\pi(s)} \mathbb{E}_{d^\pi(s)} H(O|S = s) + H(d^\pi(s)) \tag{26}$$

$$\text{s.t. } d^\pi(\{s : H(O|S = s) = \min_s H(O|S = s)\}) = 1 \tag{27}$$

For any state marginal satisfying the constraint we have:

$$\mathbb{E}_{d^\pi(s)} H(O|S=s) = C \tag{28}$$

where $C$ is a constant. Therefore, with probability 1, maximizing this objective is equivalent to maximizing the state marginal entropy in the set of dark rooms which form a $T$-cover of the state space by assumption 3(a). Therefore the Markov chain induced by the game $T$-covers the state space. Indeed, suppose by contrapositon that this is not the case. That is, there is $s$ such that for any $s'$ we have:

$$d^\pi(s') = 0 \vee d(s, s') > T \tag{29}$$

Since the set of dark rooms is a T-cover by assumption 3(a), we know that there is a dark room $s''$ such that $d(s, s'') \leq T$, which implies that $d^\pi(s'') = 0$. Therefore the state marginal entropy in the set of dark rooms is not maximized and the objective is not optimized.

□

# B  ADDITIONAL EXPERIMENTAL RESULTS

## B.1  EMERGENCE OF COMPLEXITY

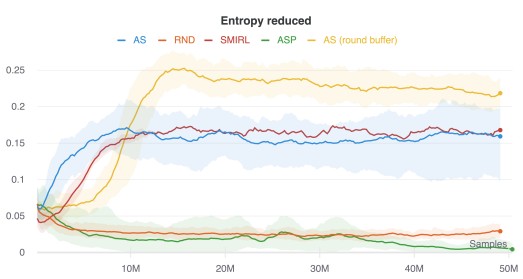

Figure 6: **Q2. Emergence of Complexity:** the average number of times the agent flip a switch to stop lights from flashing. ASP and RND do not learn to press the switch, while SMiRL and AS both press the switch a similar number of times. Resetting the AS buffer more frequently enables it to exceed even SMiRL in taking actions to control the environment.

To show that Adversarial Surprise leads to emergence of complexity by phases, we plot the temporal acquisition of two behaviors in the MiniGrid environment. The results are shown in Figure 7. This environment includes a dark room and a noisy room separated by a door. The position of the door changes at each episode. Initially, the agent is inside the noisy room and the door is open. One episode consists of 96 steps: the Control policy takes control of the agent during 32 steps, then the Explore policy takes control of the agent during 32 steps, finally the Control policy takes control of the agent during 32 steps. The first acquired behavior by the Control policy is identifying where the door is and going to the dark room during the first round. It is a *short-term* surprise minimizing behavior and an agent trained with a SMIRL objective can converge to it. However, the Explore policy learns to go back to the noisy room and to reach the farthest point from the door such that the Control policy does not have the time to reach a state of minimum entropy before the surprise of the agent is computed in the reward. This in turn incentivizes the acquisition of a more complex behavior by the Control policy: it learns to go in the dark room and to lock the agent inside by closing the door during the first round, making it harder for the Explore policy to learn to reach a state that will surprise the agent during the Control policy's second round. This behavior reminiscent of Dr. Jekyll and Mr. Hyde highlights the potential of Adversarial Surprise to learn *long-term* surprise-minimizing behaviors.

We also investigate the behaviors learned in a similar environment, which contains a switch that enables the agent to stop stochastic elements from changing color. Figure 6 measures how often agents trained with different techniques learn to press the switch. Since RND has no incentive to learn to control the environment, it never learns to press the switch. A similar result is observed for ASP, since reducing the entropy would make it easier for the Bob agent to replicate the Alice agent's final state. Thus, ASP will not always lead the agent to learn all possible behaviors relevant to controlling the environment. Both SMiRL and AS learn to take actions to reduce entropy. However, when we train AS by resetting the buffer $\beta$ used to fit the density model $p_\theta(o)$ after each *round* (that is, after both the Explore and Control policy have taken one turn), rather than after each episode, we see that AS increases the number of actions it takes to reduce entropy even over SMiRL. This is likely because resetting the buffer removes any incentive to return to states that the agent has previously seen within its lifetime, and instead gives a stronger incentive to reduce entropy immediately.

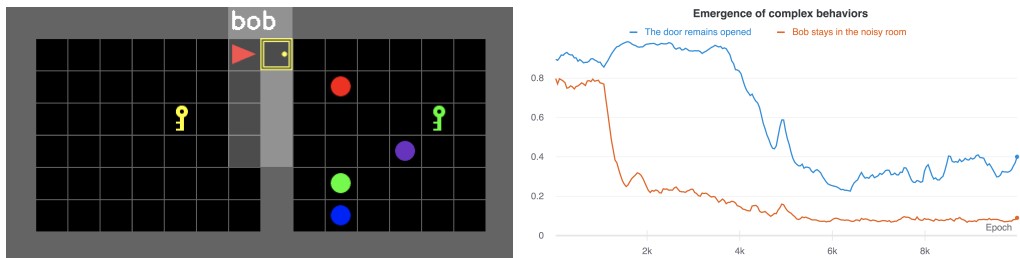

(a) The Control policy learns to lock the agent in the dark room to minimize long-term surprise.

(b) Acquisition of two behaviors by phases in order of complexity and long-term impact on the surprise.

Figure 7: **Q3. Emergence of Complexity:** a) An example environment which contains a key that can be used to lock a door separating the agent from stochastic elements. b) We observe two relatively short phase transitions separating three learning phases with three clearly distinguishable behaviors: randomly exploring, going to the dark room, locking the agent in the dark room (left). This is evidence of an emergent curriculum induced by the multi-agent competition.

## B.2    ABLATION STUDY ON THE EXPLORER AND CONTROLLER HORIZON

We perform an ablation study on the length of each round for the Explore and Control policy in the MiniGrid environment and observe that the optimal horizon is between 8 and 16.

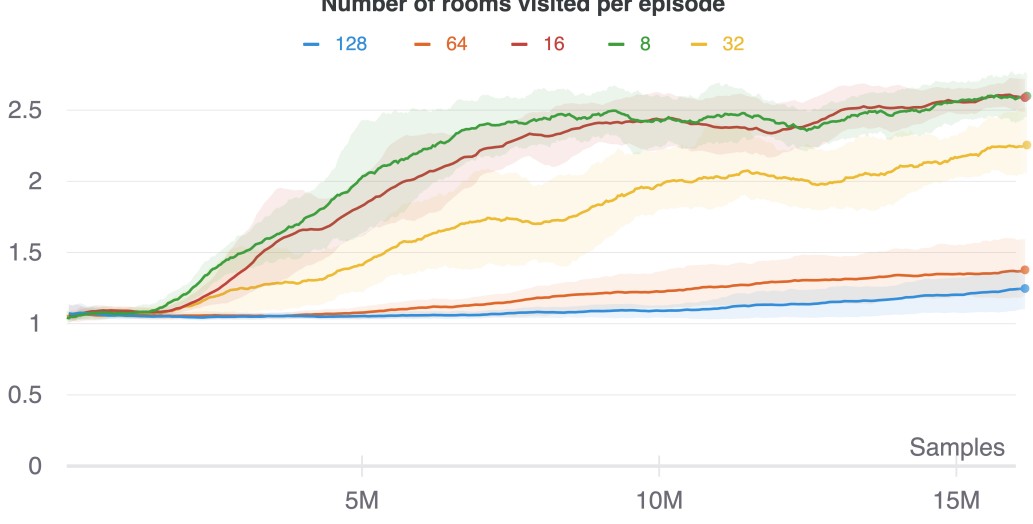

Figure 8: Ablation Study on the Explorer and Controller Horizon.

## C    HYPERPARAMETER DETAILS

In every experiment, both the explorer's and controller's policies receive a stack of the 4 last observations, a sufficient statistic of the current observation density model and the index of the current time step. The 4 last observations are stacked with the sufficient statistic before being fed into a convolutional layer. Then the output of the convolutional layer is stacked with the index of the current time step before being fed into a feed forward layer. At every step of the second-half of the controller's trajectories, we compute the error of the current observation with respect to the current observation density model and then feed the adversarial surprise buffer with the current observation to update the density model. The negative error is the reward of the controller for the current time step. By default, buffer is reset at the beginning of every episode. We also experiment with when to reset the buffer $\beta$; we find that resetting the buffer after each *round* (after the Explore policy and Control policy each take one turn) can sometimes improve performance. In our *round buffer* variant, used in 6, the buffer is reset at every round.

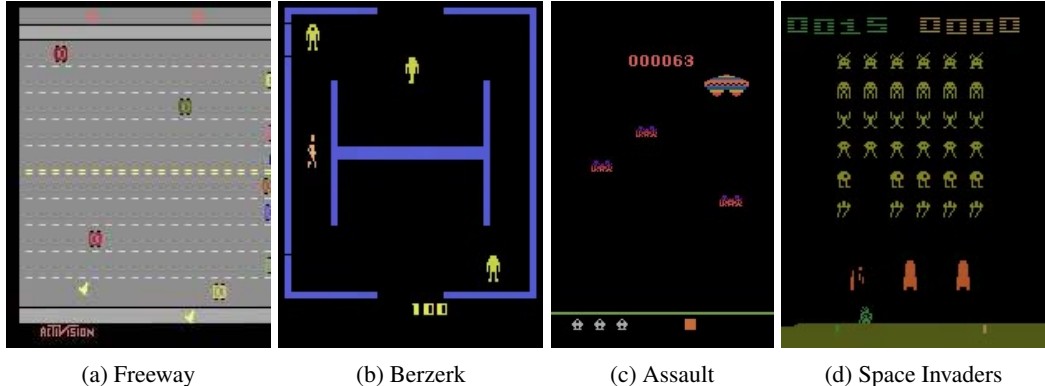

| (a) Freeway | (b) Berzerk | (c) Assault | (d) Space Invaders |

Figure 9: The four Atari environments we used to test each method.

## C.1 MINIGRID

We use 3 convolutional layers with 16, 32 and 64 output channels and a stride of 2 for every layers. For the MiniGrid experiments using $7 \times 7$ observations, for our observation density model we use $7 \times 7$ independent categorical distributions with 12 classes each instead of independent Gaussian distributions, which significantly improve the results. The sufficient statistic is the set of $7 \times 7 \times 12$ probabilities. In one episode, each agent acts during 2 rounds of 32 steps per round (Explorer-Controller-Explorer-Controller), for a total of 128 steps. We use 16 parallel environments for AS, RND, ASP and SMIRL. For the baselines, we base our RND implementation on the github repo from jcwleo, which is a reliable PyTorch implementation (https://github.com/jcwleo/random-network-distillation-pytorch), our ASP implementation on the github repo from the author (https://github.com/tesatory/hsp), our SMIRL implementation on the github repo from the author (https://github.com/Neo-X/SMiRL_Code), and our AGAC implementation is based on the Pytorch version of the github repo from the author (https://github.com/yfletberliac/adversarially-guided-actor-critic/tree/main/agac_torch.

## C.2 ATARI

We use the Atari pre-processing wrappers from Dhariwal et al. (2017) before feeding input into a five-layer architecture with three convolutional layers with 4, 32, and 64 output channels, kernel sizes of 8,4,3, and strides of 4, 2, and 1 for the explorer. We downsize the input images to $4 \times 20 \times 20$ and for our observation density model we use $4 \times 20 \times 20$ independent Gaussian distributions. The sufficient statistic is the set of means and standard deviations of the $4 \times 20 \times 20$ Gaussian distributions. For the Atari environments, we run the explorer for 64 steps and controller for 128 steps and alternate until the end of an episode and reset the buffer after every life lost. The four Atari environments used for testing are shown in Figure 9

## C.3 DOOM

We use the *Take Cover* scenario on the ViZDoom Kempka et al. (2016a) platform. We feed the input to a five-layer architecture with three convolutional layers with 32, 32, and 64 output channels, kernel sizes of 4,4,3, and strides of 2, 2, and 1 for the explorer. We downsize the input images to $4 \times 20 \times 20$ and for our observation density model we use $4 \times 20 \times 20$ independent Gaussian distributions. The sufficient statistic is the set of means and standard deviations of the $4 \times 20 \times 20$ Gaussian distributions. We similarly run the explorer for 64 steps and controller for 128 steps and alternate until the end of an episode and reset the buffer after a life has been lost. The scenario used for testing is shown in Figure 5

## C.4 PLOTS

For each environment and each method, we run a total of six random seeds and compare the top three, as a measure of the best-case performance. Plots are generated by smoothing the obtained

returns using a rolling window of steps corresponding to 25, 15, 10, 30 for Freeway, 30, 40, 30, 20 for Berzerk, 10,20,5,1 for Assault, and 10, 15, 10, 2 for Space Invaders. Then, we plot both the mean and error bars representing a 95% confidence interval over the seeds.

A website with videos of the experiments is available here: `https://sites.google.com/view/adversarial-surprise/home`.

# D   ALGORITHM

---

**Algorithm 1:** Adversarial Surprise

---

Randomly initialize $\phi^E$ and $\phi^C$;
**for** *episode* = 0, ..., M **do**
    Initialize $\theta, R^i = 0, \beta \leftarrow \{\}$, *explore_turn* = True, $t^C = k, s_0 \sim p(s_0), o_0 \sim p(O_0|s_0)$;
    **for** $t \leftarrow 0$ **to** $T$ **do**
        **if** *explore_turn* **then**
            $a_t \sim \pi^E(o_t, h_t^E)$ ;                                   // Explore
        **else**
            $a_t \sim \pi^C(o_t, h_t^C)$ ;                                   // Control
        **end**
        $s_{t+1} \sim \mathcal{T}(s_{t+1}|s_t, a_t), o_{t+1} \sim p(O_{t+1}|s_{t+1})$ ;          // Environment step
        $r_t^i = \log p_\theta(o_{t+1})$ ;                        // Compute intrinsic reward
        **if** *not explore_turn* and $t - t^C > k/2$ **then**
            $R^i = R^i + r^i$ ;
        **end**
        $\beta = \beta \cup \{o_t, a_t, o_{t+1}\}$ ;                          // Update buffer
        **if** $t == t^C$ **then**
            *explore_turn* = not *explore_turn* ;                    // Switch turns
            $t^C = t^C + 2k$;
        **end**
        $\theta_{t+1} =$MLE_update$(\beta, \theta_t)$ ;                         // Fit density model
    **end**
    $\phi^E =$RL_update$(\beta, -R^i)$ ;          // Train Explore policy $\pi^E$ with reward $-R^i$
    $\phi^C =$RL_update$(\beta, R^i)$ ;            // Train Control policy $\pi^C$ with reward $R^i$
**end**

---

# E   BROADER IMPACT:

AS is an intrinsic motivation method aimed at enhancing learning in intelligent agents. There has been a rich history of previous work in this area, where the goal has been to develop more general algorithms for learning in the absence of task-specific rewards. This work is in line with previous work and should not directly cause broader harms to society. We are not using any dataset or tool that will perpetuate or enforce bias. Nor does our method make judgements on our behalf that could be misaligned with our values. A potential positive impact of this work may be providing interesting insight into the nature of intelligence and learning.

