# OpenReview forum: "Explore and Control with Adversarial Surprise"
_ICLR.cc/2022/Conference — ICLR 2022 Submitted_

### Official Review · Reviewer_gVM6 · 2021-10-23

**Correctness:** 4
**Technical Novelty And Significance:** 3
**Empirical Novelty And Significance:** 3
**Recommendation:** 5
**Confidence:** 4

**Main Review:**

AFTER REBUTTAL

After having read the authors response, as well as other reviews and discussions, I am keeping my original, slightly negative, score.

Anyway, I would like to encourage the authors on improving the submitted draft, especially
- considering a critical evaluation of the state density estimation performance, and comparing their results with non-parametric entropy estimation,
- clarifying how subsequent RL tasks can benefit from the pre-trained models (other than zero-shot transfer).
I believe that with these improvements this work will be a nice contribution to the unsupervised RL community.

-------

While the proposed methodology is certainly interesting, my main concerns about the current version of this work regard (i) the applicability of observation density estimation and (ii) the robustness and completeness of the experimental evaluation. I report below some detailed comments on this matter.

DENSITY ESTIMATION

1) From my understanding, the method heavily relies on direct estimation of the observation marginal density to compute the objective function. However, direct density estimation is known to scale badly to the dimensionality of the space (e.g., Beirlant et al., Nonparametric entropy estimation: An overview, 1997). Moreover, density estimation does not seem to be required by the method, which just need an estimate of the observation entropy. The recent works (Mutti et al., 2021; Seo et al., 2021; Liu and Abbeel, 2021) showed that non-parametric entropy estimation can be much better in this setting. On the one hand, I would like to understand if density estimation is really working in the reported experiments (such as comparing with an estimate obtained from a huge number of samples). On the other hand, I am wondering if these results could be improved with non-parametric entropy estimation.

EMPIRICAL EVALUATION

2) In Appendix C.4, the paper reports that all the experiments are obtained with six independent runs. This number is not ideal, but reasonable in the reported settings. However, I do not understand why the plots are just showing the average (and confidence intervals) computed over the top-three seeds. The Appendix motivate this as a comparison of the best-case performance, but worst-case performance arguably matters as well. Moreover, can the authors clarify the strategy behind the hyper-parameters selection, and if they experienced any particular sensitivity to the tried values?

3) The experiments do not include a comparison with maximum state entropy methods, especially APT (Liu and Abbeel, 2021) and RE3 (Seo et al., 2021) that have been experimented in analogous domains. Moreover, an observation coverage metric is only reported for the MiniGrid experiment.

4) The experiments do not include any continuous control domain, which might represent a worst-case setting for AS. Previous works have been extensively analyzed in these kind of domains, such as Mujoco tasks in (Laskin et al., URLB: Unsupervised reinforcement learning benchmark, 2021). Note that the latter is very recent and at most concurrent with this work, but it might provide a useful inspiration.

5) I am not completely convinced about the fairness of the zero-shot transfer comparison. On the one hand, the reported results are somewhat confusing: If the setting is zero-shot, why the plots report training steps? They refer to the training steps of the unsupervised exploration phase? On the other hand, I do not clearly understand why AS should actually work in zero-shot transfer, and the reported baselines are not designed for this setting (especially RND). I think that including a comparison over a (more common) fine-tuning to downstream tasks setting would strengthen the analysis.

6) If my previous hypothesis over the interpretation of the zero-shot setting is correct, I am wondering if this couldn't be seen as a partially negative result on the stability of the method. Indeed, the performance of AS is not monotonically improving, which casts some doubts on whether the method is really converging to something that is useful to downstream tasks, or it is only effective in covering the observation space over the process.

GAME-THEORETIC ANALYSIS

7) This is a lesser concern, but I think that the paper somewhat misses the opportunity to analyze the method from a game-theoretic perspective. The min-max game optimization problem is well-posed and actually feasible? What is the solution concept one should aim to achieve in this formulation?

MINOR
- It is sometimes quite hard to inspect the plots, even when zooming on the pdf;
- Figure 2: changing colors between the two plots does not help tracking the curves, while there is some mess in the legend of Fig. 2c, and it is not clear which one refer to the SMiRL curve;
- It is of course hard to fit everything into a 9-pages paper, but I would suggest the authors to try to squeeze the figures on the "emergence of complexity" into the main text.

**Summary Of The Paper:**

This paper introduces a method, called Adversarial Surprise (AS), for unsupervised reinforcement learning. AS employs a two-player, adversarial, sequential procedure in which an Explore player tries to maximize the approximate entropy of the observations, whereas a Control player tries to minimize this same entropy. The method is then compared against other unsupervised RL baselines in visual domains, over both state coverage and zero-shot adaptation.

**Summary Of The Review:**

This paper introduces some interesting ideas. Especially, it provides a practical and smoother reinterpretation of asynchronous self-play (Sukhbaatar et al., 2017), a method holding compelling premises but with some clear limitations. Moreover, it tries to unify under a unique approach the pros of novelty-seeking exploration and surprise minimization, while mitigating some of their cons (such as noisy-tv and dark-room problems respectively). Overall, the proposed methodology seems valuable, and I believe it can help improving the state of the art in some specific settings (e.g., partially observable visual domains). Those might be sufficient reason for accepting the paper.

However, I think that the paper has some important limitations as it is, which might significantly reduce its potential impact. Thus, I am currently providing a slightly negative evaluation. I detailed in the main review above some of the concerns that the authors might address in their response.

---

> ### Author Response · Authors · 2021-11-14
> **Thank you for your insightful feedback.**
>
> We appreciate your suggestion regarding non-parametric entropy estimation. We have indeed considered replacing the density model with a non-parametric entropy estimator at several stages of the project and have decided to keep the density model because it works well in our experiments. We will test a non-parametric entropy estimator method and update you with the results.
>
> It seems that your main concern with the paper is the thoroughness of the empirical evaluation, and in particular, the use of top 3 seeds for evaluation.  In fact, our experiments were conducted using 5 random seeds, and plots show the mean and 95% CI over all 5 seeds, as stated in the caption of Figure 4. The statement in the appendix that experiments use the top 3 seeds is erroneous, and is a holdover from a previous workshop submission that we forgot to change. We have corrected this error and uploaded a revised version of the paper.
>
> To address your other concerns:
> > coverage metric is only reported for the MiniGrid experiment.
>
> Figure 3 shows state coverage in Doom.
> > If the setting is zero-shot, why the plots report training steps? They refer to the training steps of the unsupervised exploration phase?
>
> Yes, we plot an evaluation curve throughout training. At each point on the plot, after the model has trained for X steps, we take the current model and test it in the transfer environment. The model is never trained using rewards from the transfer setting, which is why we use the term “zero-shot”.
> We chose our baselines as a fair comparison to use in the no-reward setting, since ¾ of the baselines present no-reward experiments in the original papers. Indeed, AGAC, SMiRL, and ASP all conduct experiments in the no-reward setting.
>
> We have corrected the issues with Figure 2 and uploaded a revised version of the pdf.

---

> > ### Comment · Reviewer_gVM6 · 2021-11-18
> > **Thank you for your response**
> >
> > I would like to thank the authors for their replies, which are addressing some of my concerns, especially regarding the evaluation protocol: It is reassuring to see that the reported results are not coming from a top 3 seeds evaluation. I am also looking forward to see their comparison between state density estimation and non-parametric entropy estimators.
> >
> > However, I still have some concerns about the empirical evaluation of their method, which in my opinion does too little to clarify what are the factors that make AS work in the reported domains, whether their methodology would generalize to different settings, how the pre-trained model should be employed in RL exactly.
> >
> > I noticed that other reviewers have similar concerns about the experimental analysis, I can especially relate with this comment of Reviewer w5NP
> > > Some of the considered baselines were proposed or used to improve exploration in the presence of extrinsic reward (RND, AGAC, SMiRL, ASP). The outputs of other unsupervised methods such as DIAYN, VISR or APT can be used for transfer via Hierarchical RL, Generalized Policy Improvement or fine-tuning. After reading the paper, I'm unclear about how one would use AS to either aid with exploration in the presence of extrinsic reward or how the resulting policies could be used for transfer to downstream tasks. Given that the method produces two sets of neural network weights (one per policy), and the fact that the two policies probably need to be interleaved in order to produce meaningful behavior, it is not obvious how to best leverage them which in my opinion reduces the significance of the results.
> >
> > Some of my comments that the authors might consider answering in more details are:
> >
> > > can the authors clarify the strategy behind the hyper-parameters selection, and if they experienced any particular sensitivity to the tried values?
> >
> > > The experiments do not include a comparison with maximum state entropy methods, especially APT (Liu and Abbeel, 2021) and RE3 (Seo et al., 2021) that have been experimented in analogous domains.
> >
> > > I do not clearly understand why AS should actually work in zero-shot transfer, and the reported baselines are not designed for this setting (especially RND).
> >
> > > the performance of AS is not monotonically improving, which casts some doubts on whether the method is really converging to something that is useful to downstream tasks, or it is only effective in covering the observation space over the process.

---

### Official Review · Reviewer_w5NP · 2021-10-25

**Correctness:** 3
**Technical Novelty And Significance:** 3
**Empirical Novelty And Significance:** 2
**Recommendation:** 3
**Confidence:** 4

**Main Review:**

**Positives**

+ The paper is well written and generally easy to follow and understand, and the related work properly puts the proposed method in context.
+ Section 4.1 provides an interesting formalization of insights that had been experimentally observed in previous works (e.g. noisy TVs, dark rooms) that had not been mathematically described before as far as I know.
+ The method combines the advantages of surprise maximizing and surprise minimizing strategies in a conceptually simple manner which should be easy to replicate.
+ Multiple baselines are reported for each experiment.


**Concerns**

- The paper uses three metrics to evaluate unsupervised RL methods: coverage (number of rooms in MiniGrid, binned x positions in VizDoom), amount of task reward collected (VizDoom, Atari) and qualitative evaluations based on a few videos available in the project site (at the time of reviewing, videos for MiniGrid and the four Atari games were available). In my opinion, while these metrics can provide some insight on the type of behaviors AS may discover, more thorough quantitative evaluation is required in order to properly assess the usefulness of the method. For instance, a common practice in related works consists in evaluating how the output of the unsupervised pre-training stage helps when solving downstream tasks (i.e. reward functions) in the same environment.
- Some of the considered baselines were proposed or used to improve exploration in the presence of extrinsic reward (RND, AGAC, SMiRL, ASP). The outputs of other unsupervised methods such as DIAYN, VISR or APT can be used for transfer via Hierarchical RL, Generalized Policy Improvement or fine-tuning. After reading the paper, I'm unclear about how one would use AS to either aid with exploration in the presence of extrinsic reward or how the resulting policies could be used for transfer to downstream tasks. Given that the method produces two sets of neural network weights (one per policy), and the fact that the two policies probably need to be interleaved in order to produce meaningful behavior, it is not obvious how to best leverage them which in my opinion reduces the significance of the results.
- I have some concerns (and doubts) about the evaluation of unsupervised agents. Appendix C.4 states that plots report the top-3 seeds (out of 6), whereas Figure 4's caption states that 5 random seeds without any kind of filtering are reported. Which method was used in practice? I am particularly concerned about the former, as it would leak information about the task reward via seed selection.
- The choice of Atari environments is never explained, and I wonder why authors decided to run experiments on this particular subset of games. I believe that it would be much more informative if these methods had been evaluated on some of the 6 hard exploration games identified by Bellemare et al. (NeurIPS, 2016) (cited in the paper): Freeway, Gravitar, Montezuma's Revenge, Pitfall, Private Eye, Solaris and Venture.
- The experimental setting seems to vary across environments and even plots. For instance, Figure 4 shows a different number of train steps across Atari games (ranging from 180k to >400k). Authors report over 120k train steps when showcasing the advantages of AS in terms of zero-shot performance in VizDoom (Figure 5), whereas Figure 3 shows results after only 1k train steps when reporting coverage in this same environment. I would advise authors to unify the evaluation protocol for each domain (i.e. same protocol for all Atari games, same protocol for all VizDoom experiments, etc). Moreover, the term 'train step' is not always clear in RL and I was not able to find its definition in the manuscript.


**Other comments**

I have other suggestions for the authors that did not play an important role in my decision but I believe would improve the overall quality of the paper:

- Many environments feature *irreversible decisions*, i.e. actions that make the agent transition from state $s$ to $s'$ but make it impossible to go back to state $s$. I know that it is difficult to derive theoretical results for these complex environments, but I suggest being more explicit about the limitations of the theory (especially since it seems that even some of the MiniGrid environments have this property, e.g. the ones where the agent can lock a door).
- The paper mentions that $k^E$ and $k^C$ are independent hyperparameters that can be set to different values. Is AS sensitive to this choice? It would be interesting to include an analysis of the role played by these values in the discovered behaviors.
- The structure of Section 5 (questions, description about the experimental setup, answers) makes it somewhat hard to read. I would suggest something like *experimental setup, Q1 (question + answer), Q2 (question + answer), Q3 (question + answer)* instead.
- In Figure 3's caption, I would suggest using $\log p(x)$ instead of $\log p(s)$, as the true state is unknown and will probably include more information than just the agent's x position.
- In page 9, authors mention that the performance of RND decreases over time. Please see Campos et al. in the provided references for an example where using reward normalization enables very long unsupervised pre-training with RND without performance collapse.
- The text in some figures is too small (2b, 2c, 4, 5) or cropped (4a, 4b).
- The legend in Figure 2c has repeated elements. Figures 2b and 2c do not include the same baselines (e.g. SMiRL is missing in 2c). Please adjust their x limits to be the same, and adjust the y limit to avoid leaving empty half of the figure.

**Additional references**

Empowerment

> Karol Gregor, Danilo Jimenez Rezende, and Daan Wierstra. Variational intrinsic control. arXiv preprint arXiv:1611.07507, 2016.

Unsupervised training based on empowerment + fast adaptation via GPI
> Steven Hansen, Will Dabney, Andre Barreto, Tom Van de Wiele, David Warde-Farley, and Volodymyr Mnih. Fast task inference with variational intrinsic successor features. In ICLR, 2020.

Intrinsic motivation based on episodic curiosity
> Adrià Puigdomènech Badia, Pablo Sprechmann, Alex Vitvitskyi, Daniel Guo, Bilal Piot, Steven Kapturowski, Olivier Tieleman, Martín Arjovsky, Alexander Pritzel, Andew Bolt, et al. Never give up: Learning directed exploration strategies. In ICLR, 2020.
>
> Adrià Puigdomènech Badia, Bilal Piot, Steven Kapturowski, Pablo Sprechmann, Alex Vitvitskyi, Daniel Guo, and Charles Blundell. Agent57: Outperforming the atari human benchmark. In ICML, 2020.

Unsupervised pre-training with NGU/RND + transfer to downstream tasks
> Víctor Campos, Pablo Sprechmann, Steven Stenberg Hansen, Andre Barreto, Steven Kapturowski, Alex Vitvitskyi, Adria Puigdomenech Badia, and Charles Blundell. Beyond fine-tuning: Transferring behavior in reinforcement learning. In ICML 2021 Workshop on Unsupervised Reinforcement Learning. 2021.

**Summary Of The Paper:**

This paper proposes Adversarial Surprise (AS), a method for unsupervised training of RL agents based on the competition of two policies dubbed Explore and Control. These two policies control compete to maximize/minimize surprise, respectively, by taking turns to control a shared body. Authors show that under some conditions this objective leads to maximizing state entropy within an episode, and formalize the settings where other approaches may fail. Evaluation is performed on three different domains: procedurally generated MiniGrid tasks, VizDoom and four Atari games. AS explores more rooms within an episode in Minigrid, visits more x positions within an episode in VizDoom, and obtains higher zero-shot scores in 3/4 Atari games.

**Summary Of The Review:**

The paper presents an interesting method, but the evaluation protocol needs to be improved in order to properly assess the advantages of the proposed approach.

---

> ### Author Response · Authors · 2021-11-14
> **Thank you for your detailed feedback.**
>
> It seems that your main concern is with the experimental evaluation, and in particular: 1) the use of top 3 seeds, 2) evaluation on downstream tasks, and  3) the choice of environments and baselines. To address these:
> - Random seeds: We would like to correct an important misconception. Our experiments were conducted using 5 random seeds, and plots show the mean and 95% CI over all 5 seeds, as stated in the caption of Figure 4. The statement in the appendix that experiments use the top 3 seeds is erroneous, and a holdover from a previous workshop submission that we forgot to change. We have corrected this error and uploaded a revised version of the paper.
> - Evaluation on downstream tasks: You asked about metrics which explore the usefulness of an unsupervised RL method for solving downstream tasks. Figures 4 and 5 assess the performance of agents trained unsupervised with each method, when transferred to the task of obtaining game reward in Atari and Zoom. These plots focus on the evaluation performance of the agent throughout training when asked to solve downstream tasks (i.e. maximize the game reward functions) in the same environment.  In terms of how we applied AS to the downstream task, we continue to use both the Explore and the Control policy, which act for the same number of timesteps as during training.
> - Choice of environments and baselines: We chose our baselines as a fair comparison to use in the no-reward setting, since ¾ of the baselines present no-reward experiments in the original papers. AGAC, SMiRL, and ASP all conduct experiments with no rewards. RND does not, but RND is able to outperform ICM (https://arxiv.org/abs/1705.05363), which is designed to work in the no-reward setting and presents no-reward experiments.
>
> Given these baselines, we tried to choose a representative sample of environments. We do include Freeway, as you suggest. We did not include Montezuma’s revenge, because it was not explored by any of the prior no-reward baselines (AGAC, SMiRL, or ASP), which mainly focus on gridworlds, limited Atari environments, and Doom. We will begin conducting additional experiments in the environments you suggest, and update you when we have the results.
>
> In addition, we have included an analysis of the effect of the k parameter in the revised pdf, which shows that several values of k give good performance, but if k is too large, performance can suffer
>
> Thank you for your detailed comments suggesting additional citations and corrections of typos. We have made these corrections and uploaded a revised version of the pdf.

---

> > ### Comment · Reviewer_w5NP · 2021-11-15
> > **Discussion**
> >
> > Thanks for clarifying the evaluation criteria and addressing some of my comments. I'm looking forward to seeing results on harder exploration games.
> >
> > I just wanted to flag that authors have not replied to one of my main concerns: the utility of the outputs of unsupervised training with AS. Zero-shot transfer is a very good evaluation criteria for agents whose behavior is conditioned on some context, as one can evaluate whether they can generalize to new tasks or situations at test time (e.g. encoding tasks in the prompt for GPT-3, or passing previously unseen goals to goal-conditioned RL agents). However, for agents that are not goal-conditioned like the ones in this work, zero-shot transfer becomes a measure of the alignment between the intrinsic and extrinsic rewards. While in some Atari games the score can be used as a proxy to measure the exploration capabilities of agents, the usefulness of a purely exploratory agent still needs to be demonstrated quantitatively on some downstream task. In particular, these are the relevant points in my review that remain unanswered:
> >
> > > The paper uses three metrics to evaluate unsupervised RL methods: coverage (number of rooms in MiniGrid, binned x positions in VizDoom), amount of task reward collected (VizDoom, Atari) and qualitative evaluations based on a few videos available in the project site (at the time of reviewing, videos for MiniGrid and the four Atari games were available). In my opinion, while these metrics can provide some insight on the type of behaviors AS may discover, more thorough quantitative evaluation is required in order to properly assess the usefulness of the method. For instance, a common practice in related works consists in evaluating how the output of the unsupervised pre-training stage helps when solving downstream tasks (i.e. reward functions) in the same environment.
> >
> > > Some of the considered baselines were proposed or used to improve exploration in the presence of extrinsic reward (RND, AGAC, SMiRL, ASP). The outputs of other unsupervised methods such as DIAYN, VISR or APT can be used for transfer via Hierarchical RL, Generalized Policy Improvement or fine-tuning. After reading the paper, I'm unclear about how one would use AS to either aid with exploration in the presence of extrinsic reward or how the resulting policies could be used for transfer to downstream tasks. Given that the method produces two sets of neural network weights (one per policy), and the fact that the two policies probably need to be interleaved in order to produce meaningful behavior, it is not obvious how to best leverage them which in my opinion reduces the significance of the results.

---

> > > ### Author Response · Authors · 2021-11-18
> > > **Continued discussion**
> > >
> > > It is a reasonable criticism that our method, and other methods that are performing unsupervised reinforcement learning, show an evaluation that is a comparison of how well the unsupervised reinforcement learning method aligns with the original extrinsic reward of the MDP. However, one of the main goals of unsupervised reinforcement learning is to find general-purpose reward functions that will provide benefits towards training on any particular task or MDP. In an effort to show this benefit in our paper we evaluate across a number of different environments including Atari and Vizdoom to show the breadth of possible behaviours and MDP's that pre-training with AS can assist with. This type of analysis has been used in other unsupervised reinforcement learning papers [1]. Given the discussion, it's not completely clear why the type of analysis performed in the paper is insufficient and how performing a type of training where the rewards are always combined between AS and an extrinsic reward is meaningfully different from our analysis.
> > >
> > > - [1] Liu H, Abbeel P. Behavior from the void: Unsupervised active pre-training. NeurIPS 2021.

---

> > > > ### Comment · Reviewer_w5NP · 2021-11-18
> > > > **Discussion: evaluating unsupervised RL methods**
> > > >
> > > > I totally agree with authors in their understanding of what one of the main goals of unsupervised RL is:
> > > > > one of the main goals of unsupervised reinforcement learning is to find general-purpose reward functions that will provide benefits towards training on any particular task or MDP
> > > >
> > > > Indeed, this is what I was referring to in my review: there is not enough experimental evidence showing that using AS will provide such benefits on downstream tasks or MDPs. Reporting coverage over x position in VizDoom and zero-shot transfer plots in VizDoom and four Atari games does not demonstrate these benefits. Common approaches for demonstrating these benefits consist in using the method either as a pre-training strategy or as an exploration bonus, although these may not be the only options.
> > > >
> > > > In particular, authors refer to [Liu & Abbeel's work [1]](https://arxiv.org/pdf/2103.04551.pdf) as an example of a paper with a similar evaluation protocol:
> > > > >In an effort to show this benefit in our paper we evaluate across a number of different environments including Atari and Vizdoom to show the breadth of possible behaviours and MDP's that pre-training with AS can assist with. This type of analysis has been used in other unsupervised reinforcement learning papers [1].
> > > >
> > > > The last statement is incorrect. Liu & Abbeel pre-train policies using unsupervised RL, but report results after a short fine-tuning phase that shows that their pre-training strategy is indeed effective and leads to stronger results in the low-data regime. This is done both for the DeepMind Control Suite (5M unsupervised pre-training steps followed by 500k fine-tuning steps) and Atari (250M unsupervised pre-training steps followed by 100k fine-tuning steps, same protocol as in [Hansen et al.](https://arxiv.org/abs/1906.05030)). The lack of such fine-tuning phase (or any other way to leverage the outcome of the unsupervised pre-training) is what I am concerned about and what is lacking when compared to other approaches in the literature.

---

### Official Review · Reviewer_MoPi · 2021-11-02

**Correctness:** 4
**Technical Novelty And Significance:** 3
**Empirical Novelty And Significance:** 3
**Recommendation:** 5
**Confidence:** 4

**Main Review:**

Overall this is a neat idea, related to but distinct from previous works. The problem setting of a stochastic block MDP (BMDP) is well motivated although not frequently encountered in RL. The strongest results come from a new custom environment in MiniGrid, which clearly motivates the switching mechanism to discover interesting behaviors and fully explore the environment. In other environments the results are less compelling: in Doom the AGAC baseline is not included, while the Atari experiments are hard to interpret due to the choice of the four games, which is not clear. While I like the paper, I am voting for weak reject primarily because of the reporting protocol, only showing the top 3 seeds. This is known to be bad practice and does not seem justified, at least without presenting full results in the Appendix. As with any review it is possible I missed something, in which case I can change my score to reflect this during the discussion.

More Detailed Comments:

1. One of the greatest strengths of this work is that is it well motivated, and thorough. The related work section for instance positions the paper effectively within the literature, which is surprisingly rare. The flip side of this, is that the specific setting considered, stochastic BMDPs, seems very specific. It would be great to have more examples of real world settings where this setting would be present.
2. The MiniGrid experiment is a nice example of a setting where existing methods fail in stochastic environments, with a switch that allows AS to remove the stochasticity. Is it the case that the Explore agent can turn the stochasticity back on? How often does it do this?
3. Why are the methods in Fig. 2b) and 2c) different? E.g. AGAC is in one but not the other. More of an aesthetic issue but it is confusing switching the colors for methods in plots that are side by side.
4. The authors say "ViZDoom environment (Kempka et al., 2016b), which was used by AGAC and SMiRL", *but then do not include AGAC in the Doom experiments.* Why?
5. How were the four Atari games chosen? As far as I am aware, the common games used for exploration are Montezuma's Revenge and Pitfall, but these were not included. This makes the reader question the choice, for example if the method works best for these but would do equally poorly on a different set of games. While it isn't the perfect environment by any means, Montezuma's Revenge is a good benchmark because it has been predefined, so if a method does/doesn't work there it can serve as a litmus test.
6. How does AS combine with rewards? All of the other methods (aside from ASP) are designed for settings where an extrinsic reward is available, and they use this to get their best results. I know "unsupervised" or "self-supervised" RL is currently a popular topic, but in many cases we may have a reward function. Indeed - the baselines compared against were not designed to work in the regime they were tested in.
7. How do you choose k? I may have missed it but couldn't find it anywhere. The paper does mention that the ability to tune this per environment is a strength, but as a reviewer it is a concern that this may have led to the performance gains and actually the method could be very sensitive to this hyperparameter. I am thinking if k is less than the size of the room with the switch, Alice and Bob could take turns to turn it on/off and then stay in the same room forever.
8. The authors should cite RIDE, AMIGo and BeBold, three recent works studying exploration in procedurally generated environments. Comparison against them would be a nice to have but not expected, since AGAC is included and it compares against them.

Minor Comments/Typos:

* p6, reference Equation 5 but I think you mean Equation 4.
* p6, after the AS formula (min max) it says BMPD rather than BMDP. Later in that sentence, "induce"→ "induces".
* Fig 2.c) has the methods twice on the legend.
* P7 Minigrid should have a capital G.

**Summary Of The Paper:**

This paper introduces Adversarial Surprise, a new approach for unsupervised reinforcement learning in stochastic BMDPs, where the goal is to explore an environment without rewards. The algorithm uses a single agent with two policies, an Explorer and a Controller, which switch during an episode with opposite rewards: to maximize and minimize "surprise". The method is supported by a theoretical argument under the assumptions of the stochastic BMDP. When these are present in an environment, the empirical results are strong. The method is also tested in Atari and Doom.

**Summary Of The Review:**

The method is elegant and well motivated, hence a desire to see it accepted despite question marks about the experimental results. In particular, the broader applicability may be limited given the gains are clearest in the setting specifically designed to satisfy the properties which motivated the method. To increase my score I would need clarification around the evaluation protocol, which is already known to be challenging in RL without selecting the top 3 seeds (while saying 5 seeds elsewhere). I would also be happy to increase my score if there are additional baselines (e.g. AGAC for Doom) or benchmark environments (e.g. additional Atari games/MiniGrid environments), clarification around hyperparameter choices or an effective demonstration of combining AS with an extrinsic reward. Overall I could increase multiple times if this is improved.

---

> ### Author Response · Authors · 2021-11-14
> **Thank you for your thoughtful and detailed feedback.**
>
> It seems that your main concern with the paper is the thoroughness of the experimental evaluation, and in particular, the issue with using the top 3 seeds as an experiment protocol. In fact,  Our experiments were conducted using 5 random seeds, and plots show the mean and 95% CI over all 5 seeds, as stated in the caption of Figure 4. The statement in the appendix that experiments use the top 3 seeds is erroneous, and is a holdover from a previous workshop submission that we forgot to change. We have corrected this error and uploaded a revised version of the paper. Regarding your other concerns about the experimental evaluation, we address them below:
> - We apologize for the inconsistency between Figure 2 b) and c). We have corrected the plots, which are available in the revised pdf and show all baselines for each experiment.
> - We have included an analysis of the effect of the k parameter in the revised pdf, which shows that several values of k give good performance, but if k is too large, performance can suffer.
> - Regarding the Atari environments, we tried to choose a representative sample. We did not include Montezuma’s revenge, because it was not explored by any of the prior no-reward baselines (AGAC, SMiRL, or ASP), which mainly focus on gridworlds and Doom. In fact, we tried each baseline on Atari and none were able to obtain any game reward.
> - We chose our baselines as a fair comparison to use in the no-reward setting, since ¾ of the baselines present no-reward experiments in the original papers. AGAC, SMiRL, and ASP all conduct experiments with no rewards. RND does not, but RND is able to outperform ICM (https://arxiv.org/abs/1705.05363), which is designed to work in the no-reward setting and presents no-reward experiments.
> AGAC was not included in the Doom experiments because we ran out of time to conduct them. We are working on adding AGAC now and will update you with the results as soon as we can.
> - To obtain the plots showing the number of rooms explored, we have created an environment without switch (as depicted figure 2a) to clearly show that the agent does not fall in a cycle and enable better comparison with previous work. Our method also outperforms previous methods in term of exploration when switches are added.
>
> Thank you for your question about real-world examples of stochastic BMDPs. The BMDP assumption states that each observation uniquely identifies a state, but there can be many possible noisy observations of a state. We think this accurately describes many real-world settings. For example, take the case of a robot with noisy sensors navigating a home. Each image observation it receives uniquely identifies which room the robot is in (i.e. I know even from a partial image of my kitchen that I’m in the kitchen), but there are many possible noisy images it can receive from each room.
>
> Thank you for your detailed feedback regarding typos, we have corrected these mistakes in the revised version of the paper, which we uploaded. We have added citations to RIDE, AMIGo, and BeBold in the revised pdf.

---

> > ### Comment · Reviewer_MoPi · 2021-11-15
> > **Thank you for your response.**
> >
> > Hi - thank you for this!
> >
> > Some of my concerns have been addressed, for example: the seeds issue (which was significant), the additional references and adding the baselines in Fig 2. I will wait for the new experiment and the outcome of discussion with others before updating my score.
> >
> > > In fact, we tried each baseline on Atari and none were able to obtain any game reward
> >
> > Your comment on Montezuma's Revenge is interesting, since it sounds like you ran it, it might be worth including a comment in the paper that none of these methods worked. This would likely be useful for the community to know.
> >
> > > We are working on adding AGAC now and will update you with the results as soon as we can.
> >
> > I look forward to seeing it.
> >
> > In addition, it seems the paper is now above the page limit:
> > *"There will be a strict upper limit of 9 pages for the main text of the submission, with unlimited additional pages for citations. This page limit applies to both the initial and final camera ready version."*
> >
> > I am not sure how this works but presumably this needs to be fixed before the end of the discussion?

---

> > > ### Author Response · Authors · 2021-11-15
> > > **Will revise pdf**
> > >
> > > Thank you for pointing out the issue about the 9-page length. We will update the pdf to stay within 9 pages and post the result as soon as possible. We'll also add the comment about Montezuma's revenge, as requested. And we will update you about the experiments with AGAC.

---

> > > ### Author Response · Authors · 2021-11-26
> > > **Updates**
> > >
> > > Hello,
> > >
> > > We have completed the additional experiment with AGAC on Doom. We have found that In AGAC, the actor chooses actions that maximize the discrepancy between its policy and the Adversary's. Although this proves to be effective for exploration between subsequent trajectories, as shown in Figure 5, the interaction between the Control and Explore policies in Adversarial Surprise is able to facilitate more complex behaviors by not only finding novel states to explore but determining when to exhibit control over safe states.
> > > The emergence of interesting behaviour in both Atari and Vizdoom is evidenced in the videos available at: \url{https://sites.google.com/corp/view/adversarialsurprise/home}.
> > >
> > > We have also shortened the paper down to 9 pages.

---

### Decision · Program_Chairs · 2022-01-20

**Decision:**

Reject

**Comment:**

The idea of having two policies with opposing strategies, one aiming to maximize a notion of surprise whereas the other tries to minimize it, is an interesting one. However, even after the author rebuttal, all reviewers have lingering concerns about the evaluation protocol. In addition, there are remaining questions about the bonuses used; there are concerns that these only work for very specific domains. For these reasons, I'm recommending rejection. I encourage the authors to carefully read the concerns of the reviewers about evaluation and consider using a different evaluation protocol for a future version of this work.